# UNDERSTANDING GANS: THE LQG SETTING

## ABSTRACT

Generative Adversarial Networks (GANs) have become a popular method to learn a probability model from data. Many GAN architectures with different optimization metrics have been introduced recently. Instead of proposing yet another architecture, this paper aims to provide an understanding of some of the basic issues surrounding GANs. First, we propose a natural way of specifying the loss function for GANs by drawing a connection with supervised learning. Second, we shed light on the generalization peformance of GANs through the analysis of a simple LQG setting: the generator is **linear**, the loss function is **quadratic** and the data is drawn from a **Gaussian** distribution. We show that in this setting: 1) the optimal GAN solution converges to population Principal Component Analysis (PCA) as the number of training samples increases; 2) the number of samples required scales exponentially with the dimension of the data; 3) the number of samples scales almost linearly if the discriminator is constrained to be quadratic. Thus, linear generators and quadratic discriminators provide a good balance for fast learning.

## 1 INTRODUCTION

Learning a probability model from data is a fundamental problem in statistics and machine learning. Building off the success of deep learning methods, Generative Adversarial Networks (GANs) (Goodfellow et al., 2014) have given this age-old problem a face-lift. In contrast to traditional methods of parameter fitting like maximum likelihood estimation, the GAN approach views the problem as a *game* between a *generator* whose goal is to generate fake samples that are close to the real data training samples and a *discriminator* whose goal is to distinguish between the real and fake samples. The generator and the discriminators are typically implemented by deep neural networks. GANs have achieved impressive performance in several domains (e.g., (Ledig et al., 2016; Reed et al., 2016)). Since (Goodfellow et al., 2014), many variations of GANs have been developed, including $f$-GAN (Nowozin et al., 2016), MMD-GAN (Dziugaite et al., 2015; Li et al., 2015), WGAN (Arjovsky et al., 2017), improved WGAN (Gulrajani et al., 2017), relaxed WGAN (Guo et al., 2017), Least-Squares GAN (Mao et al., 2016), Boundary equilibrium GAN (Berthelot et al., 2017), etc. These GANs use different metrics in the optimization for training the generator and discriminator networks (Liu et al., 2017).

The game theoretic formulation in GANs can be viewed as the dual of an optimization that minimizes a distance measure between the empirical distributions of the fake and real samples. In the population limit where there are infinite number of samples, this optimization minimizes the distance between the generated distribution and the true distribution from which the data is drawn. In the original GAN framework, this distance measure is the Jenson Shannon divergence. However, Arjovsky et al (Arjovsky et al., 2017) noted that this distance does not depend on the generated distribution whenever its dimension is smaller than that of the true distribution. In this typical case, the Jenson Shannon divergence does not serve as a useful criterion in choosing the appropriate generated distribution. To resolve this issue, (Arjovsky et al., 2017) proposed the Wasserstein GAN (WGAN) which uses the first-order Wasserstein distance instead of Jensen-Shannon divergence. This distance is meaningful even when the dimension of the generated distribution is less than the true distribution. Nevertheless there are many other distance measures that satisfy this criterion and it is not clear how to choose among them. This is responsible in part for the fact that there are so many different GAN architectures. In fact, there is currently some confusion in the literature even on the basic issue of how to specify the loss function for GANs. For example, while the "Wasserstein" in Wasserstein

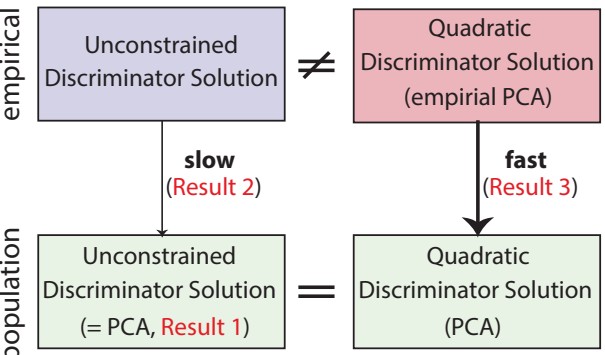

Figure 1: Summary of main results in the LQG setting. The population optimal GAN solution is PCA when the discriminator is unconstrained and when the discriminator is constrained to be quadratic. But the convergence to the population optimal is exponentially faster under a quadratic constraint on the discriminator.

GAN refers to the use of Wasserstein distance in the *distance measure* between the generated and true distributions, the "Least Squares" in Least-Squares GAN (Mao et al., 2016) refers to the use of squared error in the *discriminator optimization objective*. These are two totally different types of objects. The situation with GANs is in contrast to that in supervised learning, where how the loss function is specified in the formulation is clear and quite universally accepted.

A central issue in any learning problem is *generalization*: how close a model learnt from a finite amount of data is to the true distribution. Or, in statistical terms, how fast is the rate of convergence of the learnt model to the true model as a function of number of samples? Arora et al (Arora et al., 2017) have recently studied this problem for GANs. They observed that for Wasserstein GAN, if there are no constraints on the generator or the discriminator, the number of samples required to converge scales exponentially in the data dimension. They then showed that if the discriminator is constrained to be in a parametric family, then one can achieve convergence almost linearly in the number of parameters in that family (Theorem 3.1 in (Arora et al., 2017)). However, the convergence is *no longer* measured in the Wasserstein distance but in a new distance measure they defined (the neural network distance). The result is interesting as it highlights the role of the discriminator in generalization, but it is somewhat unsatisfactory in that the distance measure needs to be modified to tailor to the specific constraints on the discriminator. Moreover, the result requires the invention of (yet) another family of distance measures for GANs.

In this paper, we first argue that there is a natural way to specify the loss function $\ell$ for GANs, in an analogous way as in the supervised learning setup. The resulting optimal GAN minimizes a generalized loss-function dependent Wasserstein distance between the generated distribution and the true distribution, and the dual formulation of this generalized Wasserstein distance leads to a loss-function dependent discriminator architecture. To study the impact of the constraints on the generator and the discriminator on the generalization performance in this distance measure, we focus on the case when the true data distribution is Gaussian. In this case, a natural loss function to consider is quadratic, and a natural class of generators to consider is linear with a given rank $k$. In this setting, the optimal GAN minimizes the *second-order* Wasserstein distance between the generated distribution and the empirical data distribution among all linear generators of a given rank. We show the following results:

1 In the population limit as the number of data samples grow, the optimal generated distribution is the rank $k$ Gaussian distribution retaining exactly the top $k$ principal components of the true distribution, i.e. GAN performs PCA in the population limit.

2 The number of samples required for convergence in (second-order) Wasserstein distance however scales exponentially with the dimension of the data distribution.

3 Under a further constraint that the discriminator is *quadratic*, GAN converges to the same population-optimal PCA limit, but with the number of samples scaling almost linearly with

the dimension. The constrained GAN simply performs empirical PCA, and in the case when the rank $k$ of the generator is the same as the dimension of the data distribution, GAN is equivalent to maximum likelihood estimation of the underlying Gaussian model.

These results are summarized in Figure 1. The GAN architecture with a linear generator and a quadratic discriminator is shown in Figure 4.

(Arora et al., 2017) observed that the number of samples required to generalize for GAN is exponential in the dimension of the data when there are no constraints on either the generator or the discriminator. (They proved the result for first-order Wasserstein distance but a similar result holds for second-order Wasserstein distance as well, see Lemma 2 in Section 3.) Result 2 above says that even constraining the generator drastically to be linear cannot escape this exponential scaling. Result 3 says that this exponential scaling is not due to statistical limitation, but much better inference can be obtained by constraining the discriminator appropriately. Similar to Theorem 3.1 in (Arora et al., 2017), Result 3 points to the importance of constraining the discriminator. But there are two key differences. First, the convergence in Result 3 is with respect to the *original* (second-order) Wasserstein distance, not another distance measure tailored to the constraint on the discriminator. Thus, the original quadratic loss function is respected. Second, the population limit is the same as the PCA limit achieved without constraints on the discriminator. Thus, by imposing a discriminator constraint, the rate of convergence is drastically improved without sacrificing the limiting performance. There is no such guarantee in (Arora et al., 2017). Our results also provide concrete evidence that an appropriate balance between the classes of generators and discriminators, i.e. linear generators and quadratic discriminators, can provide fast training.

The Linear-Quadratic-Gaussian (LQG) setting, dating back to at least Gauss, has been widely used across many fields, including statistics, machine learning, control, signal processing and communication. It has resulted in celebrated successes such as linear regression, the Wiener filter, the Kalman filter, PCA, etc., and is often used to establish a baseline to understand more complex models. We believe it serves a similar role here for GANs[1]. Indeed it allows us to make a clear connection between GAN and PCA, perhaps the most basic of unsupervised learning methods. Moreover, even in this simple setting, the generalization issues in GAN are non-trivial, and understanding them in this setting provides the foundation to tackle more complex data distributions and more complex generators and discriminators such as deep nets.

The rest of the paper is organized as follows. In Section 2, we discuss a formulation of the GAN problem for general loss functions. In Section 3, we specialize to the LQG setting and analyze the generalization performance of GAN. In Section 4, we analyze the performance of GAN under a quadratic constraint on the discriminator. In Section 5, we present some experimental results.

## 2   A GENERAL FORMULATION FOR GANS

Let $\{\mathbf{y}_i\}_{i=1}^n$ be $n$ observed data points in $\mathbb{R}^d$ drawn i.i.d. from the distribution $\mathbb{P}_Y$. Let $\mathbb{Q}_Y^n$ be the empirical distribution of these observed samples. Moreover, let $\mathbb{P}_X$ be a normal distribution $\mathcal{N}(\mathbf{0}, \mathbf{I}_k)$. GANs can be viewed as an optimization that minimizes a distance between the observed empirical distribution $\mathbb{Q}_Y^n$ and the generated distribution $\mathbb{P}_{g(X)}$. The *population* GAN optimization replaces $\mathbb{Q}_Y^n$ with $\mathbb{P}_Y$. The question we ask in this section is: what is a natural way of specifying a loss function $\ell$ for GANs and how it determines the distance?

### 2.1   WGAN REVISITED

Let us start with the WGAN optimization (Arjovsky et al., 2017):

$$\inf_{g(.)\in\mathcal{G}} W_1(\mathbb{P}_Y, \mathbb{P}_{g(X)}), \tag{1}$$

where $\mathcal{G}$ is the set of generator functions, and the $p$-th order Wasserstein distance between distributions $\mathbb{P}_{Z_1}$ and $\mathbb{P}_{Z_2}$ is defined as (Villani, 2008)

$$W_p^p(\mathbb{P}_{Z_1}, \mathbb{P}_{Z_2}) := \inf_{\mathbb{P}_{Z_1, Z_2}} \mathbb{E}\left[\|Z_1 - Z_2\|^p\right], \tag{2}$$

---

[1]The importance of baselines in machine learning was also expressed by Ben Recht (talk at Stanford, Oct 18 2017).

where the minimization is over all joint distributions with marginals fixed. Replacing (2) in (1), the WGAN optimization can be re-written as

$$\inf_{g(.)\in\mathcal{G}} \inf_{\mathbb{P}_{g(X),Y}} \mathbb{E}\left[\|Y - g(X)\|\right]. \tag{3}$$

or equivalently:

$$\inf_{\mathbb{P}_{X,Y}} \inf_{g(.)\in\mathcal{G}} \mathbb{E}\left[\|Y - g(X)\|\right], \tag{4}$$

where the minimization is over all joint distributions $\mathbb{P}_{X,Y}$ with fixed marginals $\mathbb{P}_X$ and $\mathbb{P}_Y$.

We now connect (4) to the *supervised learning* setup. In supervised learning, the joint distribution $\mathbb{P}_{X,Y}$ is fixed and the goal is to learn a relationship between the feature variable represented by $X \in \mathbb{R}^k$, and the target variable represented by $Y \in \mathbb{R}^d$, according to the following optimization:

$$\inf_{g(.)\in\mathcal{G}} \mathbb{E}\left[\ell\left(Y, g(X)\right)\right], \tag{5}$$

where $\ell$ is the *loss* function. Assuming the marginal distribution of $X$ is the same in both optimizations (4) and (5), we can connect the two optimization problems by setting $\ell(y, y') = \|y - y'\|$ in optimization (5). Note that for every fixed $\mathbb{P}_{X,Y}$, the solution of the supervised learning problem (5) yields a predictor $g$ which is a feasible solution to the WGAN optimization problem (4). Therefore, the WGAN optimization (3) can be re-interpreted as solving the *easiest* such supervised learning problem, over all possible joint distributions $\mathbb{P}_{X,Y}$ with fixed $\mathbb{P}_X$ and $\mathbb{P}_Y$.

## 2.2 FROM SUPERVISED TO UNSUPERVISED LEARNING

GAN is a solution to an unsupervised learning problem. What we are establishing above is a general connection between supervised and unsupervised learning problems: a good predictor $g$ learnt in a supervised learning problem can be used to generate samples of the target variable Y. Hence, to solve an unsupervised learning problem for $Y$ with distribution $\mathbb{P}_Y$, one should solve the easiest supervised learning problem $\mathbb{P}_{X,Y}$ with given marginal $\mathbb{P}_Y$ (and $\mathbb{P}_X$, the randomness generating distribution). This is in contrast to the traditional view of the unsupervised learning problem as observing the feature variable $X$ without the label $Y$. (Thus in this paper we break with tradition and use $Y$ to denote data and $X$ as randomness for the generator in stating the GAN problem.)

This connection between supervised and unsupervised learning leads to a natural way of specifying the loss function in GANs: we simply replace the $\ell_2$ Euclidean norm in (3) with a general loss function $\ell$:

$$\inf_{g(.)\in\mathcal{G}} \inf_{\mathbb{P}_{g(X),Y}} \mathbb{E}\left[\ell\left(Y, g(X)\right)\right]. \tag{6}$$

The inner optimization is the optimal transport problem between distributions of $g(X)$ and $Y$ (Villani, 2008) with general cost $\ell$. This is a linear programming problem for general cost, so there is always a dual formulation (the Kantorovich dual (Villani, 2008)). The dual formulation can be interpreted as a generalized discriminator optimization problem for the cost $\ell$. (For example, in the case of $\ell$ being the Euclidean norm, we get the WGAN architecture; see Figure 2(a).) Hence, we propose (6) as a formulation of GANs for general loss functions.

## 2.3 QUADRATIC LOSS

The most widely used loss function in supervised learning is the quadratic loss: $\ell(y, y') = \|y - y'\|^2$ (*squared* Euclidean norm). Across many fields its use had led to many important discoveries. With the connection between supervised and unsupervised learning in mind, this loss function should be a prime choice to consider in GANs as well. This choice of the loss function leads to the *quadratic GAN* optimization:

$$\inf_{g(.)\in\mathcal{G}} W_2^2(\mathbb{P}_Y, \mathbb{P}_{g(X)}). \tag{7}$$

Since Wasserstein distances are weakly continuous measures in the probability space (Villani, 2008), similar to WGAN, the optimization of the quadratic GAN is well-defined even if $k < d$. The dual

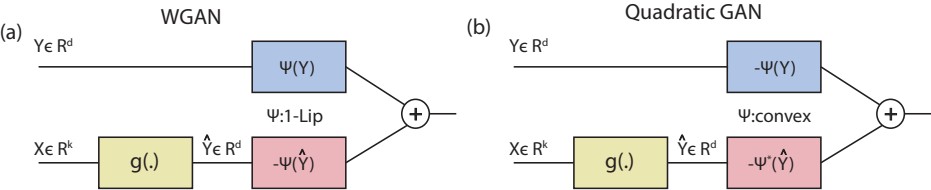

Figure 2: Dual (min-max) formulations of (a) WGAN, and (b) Quadratic GAN.

formulation (discriminator) for $W_2$ is shown in Figure 2(b). Note that in this dual, the discriminator applies $\psi$ to the real data and the convex conjugate $\psi^*$ to the generated (fake) data.

The *empirical* quadratic GAN optimization can be formulated by replacing $\mathbb{P}_Y$ with the empirical distribution $\mathbb{Q}_Y^n$ of the data as follows:

$$\inf_{g(.)\in\mathcal{G}} W_2^2(\mathbb{Q}_Y^n, \mathbb{P}_{g(X)}). \tag{8}$$

Note that while in practice one generates fake samples from $X$, we will keep the notations simpler in this paper by assuming we can generate the exact distribution $g(X)$, i.e. we can generate as many fake samples as we wish. Almost all our results can be extended to the case when we have finite number of samples from $X$ comparable to the number of samples from $Y$.

For the rest of the paper, we will focus on the problem (8) for the particular case of $Y$ Gaussian of dimension $d$, and $g$ linear of rank $k \leq d$. This is the **LQG** setting for GANs.

# 3 GANs under the LQG setup

## 3.1 The Population GAN Optimization

First, we analyze the population GAN optimization under the LQG setup. We have the following lemma:

**Lemma 1** *Let $\mathcal{S}$ be a $k$ dimensional subspace in $\mathbb{R}^d$. Let $\hat{Y}$ be a random variable whose support lies in $\mathcal{S}$. Then, $\hat{Y}^*$, the optimal solution of the optimization*

$$\inf_{\mathbb{P}_{\hat{Y}}} W_2^2(\mathbb{P}_Y, \mathbb{P}_{\hat{Y}}), \tag{9}$$

*is the projection of $Y$ to $\mathcal{S}$.*

**Proof 1** *See Appendix B.2.*

This Lemma holds even if $\mathbb{P}_Y$ is a non-Gaussian distribution. However, $\mathbb{P}_{\hat{Y}^*}$ cannot be generated as $g(X)$ when $\mathbb{P}_X \sim \mathcal{N}(\mathbf{0}, \mathbf{I}_k)$ and $g(.)$ is restricted to be linear.

Using Lemma 1 and under the LQG setup, we show that the optimal solution for the population GAN optimization is the same as the PCA solution. PCA is the most standard unsupervised learning approach (Jolliffe, 2002). PCA computes an optimal linear mapping from $Y$ to $\hat{Y}$ under the rank constraint on the covariance matrix of $\hat{Y}$ ($\mathbf{K}_{\hat{Y}}$). We say $\hat{Y}$ is the $k$-PCA solution of $Y$ if $\mathbf{K}_{\hat{Y}}$ is a rank $k$ matrix whose top $k$ eigenvalues and eigenvectors are the same as top $k$ eigenvalues and eigenvectors of the covariance matrix of $Y$ ($\mathbf{K}_Y$).

**Theorem 1** *Let $Y \sim \mathcal{N}(\mathbf{0}, \mathbf{K}_Y)$ where $\mathbf{K}_Y$ is full-rank. Let $X \sim \mathcal{N}(\mathbf{0}, \mathbf{I}_k)$ where $k \leq d$. The optimal population GAN solution of optimization (7) under linear $\mathcal{G}$ is the $k$-PCA solution of $Y$.*

**Proof 2** *See Appendix B.3.*

Lemma 1 holds if we replace $W_2$ with $W_1$. However, the conclusion of Theorem 1 is tied to the $W_2$ distance because the PCA optimization also considers the quadratic projection loss.

### 3.2 THE EMPIRICAL GAN OPTIMIZATION

In reality, one solves the GAN optimization over the empirical distribution of the data $\mathbb{Q}_Y^n$, not the population distribution $\mathbb{P}_Y$. Thus, it is important to analyze how close optimal empirical and population GAN solutions are in a given sample size $n$. This notion is captured in the generalization error of the GAN optimization, defined as follows:

**Definition 1 (Generalization of GANs)** *Let $n$ be the number of observed samples from $Y$. Let $\hat{g}(.)$ and $g^*(.)$ be the optimal generators for empirical and population GANs respectively. Then,*

$$d_{\mathcal{G}}(\mathbb{P}_Y, \mathbb{Q}_Y^n) := W_2^2(\mathbb{P}_Y, \mathbb{P}_{\hat{g}(X)}) - W_2^2(\mathbb{P}_Y, \mathbb{P}_{g^*(X)}), \tag{10}$$

*is a random variable representing the excess error of $\hat{g}$ over $g^*$, evaluated on the true distribution.*

$d_{\mathcal{G}}(\mathbb{P}_Y, \mathbb{Q}_Y^n)$ can be viewed as a distance between $\mathbb{P}_Y$ and $\mathbb{Q}_Y^n$ which depends on $\mathcal{G}$. To have a proper generalization property, one needs to have $d_{\mathcal{G}}(\mathbb{P}_Y, \mathbb{Q}_Y^n) \to 0$ quickly as $n \to \infty$. Before analyzing the convergence rate of $d_{\mathcal{G}}(\mathbb{P}_Y, \mathbb{Q}_Y^n)$ for linear $\mathcal{G}$, we characterize this rate for an unconstrained $\mathcal{G}$. For an unconstrained $\mathcal{G}$, the second term of (10) is zero (this can be seen using a space filling generator function (Cannon & Thurston, 1987)). Moreover, $\mathbb{P}_{\hat{g}(X)}$ can be arbitrarily close to $\mathbb{Q}_Y^n$. Thus, we have

**Lemma 2** *If $\mathcal{G}$ is unconstrained, we have*

$$d_{\mathcal{G}}(\mathbb{P}_Y, \mathbb{Q}_Y^n) = W_2^2(\mathbb{P}_Y, \mathbb{Q}_Y^n), \tag{11}$$

*which goes to zero with high probability with the rate of $\mathcal{O}(n^{-2/d})$.*

The approach described for the unconstrained $\mathcal{G}$ corresponds to the *memorization* of the empirical distribution $\mathbb{Q}_Y^n$ using the trained model. Note that one can write

$$n^{-\frac{2}{d}} = 2^{-\frac{2\log(n)}{d}}.$$

Thus, to have a small $W_2^2(\mathbb{P}_Y, \mathbb{Q}_Y^n)$, the number of samples $n$ should be exponentially large in $d$ (Canas & Rosasco, 2012). It is possible that for some distributions $\mathbb{P}_Y$, the convergence rate of $W_2^2(\mathbb{P}_Y, \mathbb{Q}_Y^n)$ is much faster than $\mathcal{O}(n^{-2/d})$. For example, (Weed & Bach, 2017) shows that if $\mathbb{P}_Y$ is clusterable (i.e., $Y$ lies in a fixed number of separate balls with fixed radii), then the convergence of $W_2^2(\mathbb{P}_Y, \mathbb{Q}_Y^n)$ is fast. However, even in that case, one optimal strategy would be to memorize observed samples, which does not capture what GANs do.

In supervised learning, constraining the predictor to be from a small family improves generalization. A natural question is whether constraining the family of generator functions $\mathcal{G}$ can improve the generalization of GANs. In the LQG setting, we are constraining the generators to be linear. To simplify calculations, we assume that $Y \sim \mathcal{N}(\mathbf{0}, \mathbf{I}_d)$ and $d = k$. Under these assumptions, the GAN optimization (8) can be re-written as

$$\min_{\mu, \mathbf{K}} \ W_2^2(\mathbb{Q}_Y^n, \mathcal{N}(\mu, \mathbf{K})), \tag{12}$$

where $\mathbf{K}$ is the covariance matrix with the eigen decomposition $\mathbf{K} = \mathbf{U}\mathbf{\Sigma}\mathbf{U}^t$. The optimal population solution of this optimization is $\mu_{pop}^* = \mathbf{0}$ and $\mathbf{K}_{pop}^* = \mathbf{I}$, which provides a zero Wasserstein loss.

**Theorem 2** *Let $\mu_n^*$ and $\mathbf{K}_n^*$ be optimal solutions for optimization* (12). *Then, $\mu_n^* \to \mathbf{0}$ with the rate of $\mathcal{O}(d/n)$ and $\text{Tr}(\mathbf{\Sigma}_n^*) \to d$ with the rate of $\mathcal{O}(n^{-2/d})$.*

**Proof 3** *See Appendix B.4.*

It turns out that $\text{Tr}(\mathbf{\Sigma}_n^*)$, which is a random variable, is strongly concentrated around its expectation. Thus, Theorem 2 indicates that there is a significant bias in GAN's estimation of the true distribution which translates to the slow convergence of the generalization error. Note that in the Wasserstein space, the empirical distribution $\mathbb{Q}_Y^n$ and the population distribution $\mathbb{P}_Y$ are far from each other (the distance between them concentrates around $n^{-2/d}$ (Canas & Rosasco, 2012)). Thus, if there exists another Gaussian distribution within the sphere around $\mathbb{Q}_Y^n$ with the radius of $n^{-2/d}$,

the Wasserstein-based learning method will converge to the wrong Gaussian distribution. This phenomenon causes a bias in estimating the true distribution.

Theorem 2 considers the regime where $k = d$. In practice, the dimension of the generated distribution is often much smaller than that of the true one (i.e., $k \ll d$). In this case, GAN's convergence rate can be increased from $\mathcal{O}(n^{-2/d})$ to $\mathcal{O}(n^{-2/k})$. However, this faster convergence comes at the expense of the increased bias term of the excess error (the second term of (10)). The trade-off is favorable if $Y$ is near low rank. Nevertheless even the convergence rate of $\mathcal{O}(n^{-2/k})$ is still slow. In practice, however, GANs have demonstrated impressive performance. In the next section, we show that by suitably constraining the GAN optimization, the convergence rate can be improved exponentially.

## 4 GANs with Constrained Discriminators

In this section, first we review the min-max (dual) formulation of WGAN (Arjovsky et al., 2017). Then, we characterize the min-max formulation of the quadratic GAN. Finally, we show that a properly *constrained* quadratic GAN achieves the empirical PCA solution, which converges to the population optimal with an exponentially faster rate of convergence compared to the case when the discriminator is unconstrained.

Using the Kantorovich duality (Villani, 2008), the first-order Wasserstein distance $W_1(\mathbb{P}_Y, \mathbb{P}_{g(X)})$ can be written as the following optimization (Arjovsky et al., 2017):

$$W_1(\mathbb{P}_Y, \mathbb{P}_{g(X)}) = \sup_{\psi(.):\text{1-Lip}} \mathbb{E}\left[\psi(Y) - \psi(\hat{Y})\right], \tag{13}$$

where the function $\psi(.)$ is restricted to be 1-Lipschitz. This dual formulation of $W_1$ is then used in optimization (1) to implement WGAN in a min-max architecture similar to the one of the original GAN (Figure 2). In this architecture, $\psi(.)$ is implemented by deep neural networks.

In a similar way, one can write the second-order Wasserstein distance $W_2^2(\mathbb{P}_Y, \mathbb{P}_{g(X)})$ as the following optimization (Villani, 2008):

$$W_2^2(\mathbb{P}_Y, \mathbb{P}_{g(X)}) = \mathbb{E}[\|Y\|^2] + \mathbb{E}[\|g(X)\|^2] + 2 \sup_{\psi(.):\text{convex}} - \mathbb{E}[\psi(Y)] - \mathbb{E}[\psi^*(g(X))], \tag{14}$$

where $\psi^*(\hat{y}) := \sup_{\mathbf{v}}(\mathbf{v}^t\hat{y} - \psi(\mathbf{v}))$ is the convex-conjugate of the function $\psi(.)$. Similarly, this dual formulation of $W_2^2$ can be used to implement the quadratic GAN optimization (7) in a min-max architecture which can be interpreted as a game between optimizing two functions $g(.)$ and $\psi(.)$ (Figure 2).

The following lemma characterizes the optimal solution of optimization (14) (Chernozhukov et al., 2017):

**Lemma 3** *Let $\mathbb{P}_Y$ be absolutely continuous whose support contained in a convex set in $\mathbb{R}^d$. For a fixed $g(.) \in \mathcal{G}$, let $\psi^{opt}$ be the optimal solution of optimization (14). This solution is unique. Moreover, we have*

$$\bigtriangledown \psi^{opt}(Y) \overset{dist}{=} g(X), \tag{15}$$

*where $\overset{dist}{=}$ means matching distributions.*

In the LQG setup, since $g(X)$ is Gaussian, $\bigtriangledown \psi^{opt}$ is a linear function. Thus, without loss of generality, $\psi(.)$ in the discriminator optimization can be constrained to $\psi(\mathbf{y}) = \mathbf{y}^t \mathbf{A} \mathbf{y}/2$ where $\mathbf{A}$ is positive semidefinite. Therefore, we have

$$W_2^2\left(\mathbb{P}_Y, \mathbb{P}_{g(X)}\right) = \mathbb{E}[\|Y\|^2] + \mathbb{E}[\|g(X)\|^2] + 2 \sup_{\psi(\mathbf{y})=\mathbf{y}^t\mathbf{A}\mathbf{y}/2, \mathbf{A}\succeq 0} - \mathbb{E}[\psi(Y)] - \mathbb{E}[\psi^*(g(X))]$$

$$\tag{16}$$

$$= \text{Tr}(\mathbf{K}_Y) + \text{Tr}(\mathbf{K}_{g(X)}) + \sup_{\mathbf{A}\succeq 0} - \text{Tr}(\mathbf{A}\mathbf{K}_Y) - \text{Tr}(\mathbf{A}^\dagger \mathbf{K}_{g(X)}),$$

where $\mathbf{A}^\dagger$ is the pseudo inverse of the matrix $\mathbf{A}$.

Now let $\tilde{Y}$ be a random variable whose distribution matches the empirical distribution $\mathbb{Q}_Y^n$. Similarly we can write:

$$W_2^2(\mathbb{P}_{\tilde{Y}}, \mathbb{P}_{g(X)}) = \mathbb{E}[\|\tilde{Y}\|^2] + \mathbb{E}[\|g(X)\|^2] + 2 \sup_{\psi(.):\text{convex}} -\mathbb{E}\left[\psi(\tilde{Y})\right] - \mathbb{E}[\psi^*(g(X))]. \quad (17)$$

For $W_2^2(\mathbb{P}_{\tilde{Y}}, \mathbb{P}_{g(X)})$, however, we cannot restrict $\psi$ to convex quadratic functions because $\tilde{Y}$ is a discrete variable while $g(X)$ is Gaussian. Thus, Lemma 3 implies that $\bigtriangledown\psi^{opt}$ for (17) cannot be linear. Nevertheless, by constraining to quadratic discriminators, we obtain a lower bound:

$$W_2^2(\mathbb{P}_{\tilde{Y}}, \mathbb{P}_{g(X)}) > \mathbb{E}[\|\tilde{Y}\|^2] + \mathbb{E}[\|g(X)\|^2] + 2 \sup_{\psi(\mathbf{y})=\mathbf{y}^t\mathbf{A}\mathbf{y}/2, \mathbf{A}\succeq 0} -\mathbb{E}\left[\psi(\tilde{Y})\right] - \mathbb{E}[\psi^*(g(X))] \quad (18)$$

$$= \text{Tr}(\hat{\mathbf{K}}_Y) + \text{Tr}(\mathbf{K}_{g(X)}) + \sup_{\mathbf{A}\succeq 0} -\text{Tr}(\mathbf{A}\hat{\mathbf{K}}_Y) - \text{Tr}(\mathbf{A}^\dagger\mathbf{K}_{g(X)})$$

$$= W_2^2(\mathbb{P}_Z, \mathbb{P}_{g(X)}),$$

where $\hat{\mathbf{K}}_Y = \mathbb{E}[\tilde{Y}\tilde{Y}^t]$ (the empirical covariance matrix) and $Z \sim \mathcal{N}(\mathbf{0}, \hat{\mathbf{K}}_Y)$ [2]. Therefore, the empirical constrained quadratic GAN solves the following optimization:

$$\inf_{g(.)\in\mathcal{G}} W_2^2(\mathbb{P}_Z, \mathbb{P}_{g(X)}). \quad (19)$$

Using Theorem 1, the optimal $g(X)$ to this problem is the empirical PCA solution, i.e. keeping the top $k$ principal components of the *empirical covariance matrix*.

**Theorem 3** *Under the LQG setup, the solution of the empirical constrained quadratic GAN optimization is equivalent to the empirical PCA.*

Consider the case where $d = k$ (the case $k < d$ is similar). The second term in the generalization distance $d_{\mathcal{G}}(\mathbb{P}_Y, \mathbb{Q}_Y^n)$ (10) is zero. Therefore, we have

$$d_{\mathcal{G}}(\mathbb{P}_Y, \mathbb{Q}_Y^n) = W_2^2(\mathbb{P}_Y, \mathbb{P}_Z) = W_2^2\left(\mathcal{N}(0, \mathbf{K}_Y), \mathcal{N}(0, \hat{\mathbf{K}}_Y)\right). \quad (20)$$

The $W_2^2$ distance between two Gaussians depends only on the covariance matrices. More specifically:

$$W_2^2\left(\mathcal{N}(0, \mathbf{K}_Y), \mathcal{N}(0, \hat{\mathbf{K}}_Y)\right) = \text{Tr}(\mathbf{K}_Y) + \text{Tr}(\hat{\mathbf{K}}_Y) - 2\text{Tr}\left(\left(\mathbf{K}_Y^{1/2}\hat{\mathbf{K}}_Y\mathbf{K}_Y^{1/2}\right)^{1/2}\right). \quad (21)$$

Hence, the convergence of this quantity only depends on the convergence of the empirical covariance to the population covariance, together with smoothness property of this function of the covariance matrices. The convergence has been established to be at a quick rate of $\tilde{\mathcal{O}}(\sqrt{d/n})$ (Rippl et al., 2016).

Finally, note that if one has a finite number of $X$ samples (replacing $\mathbb{P}_{g(X)}$ with $\mathbb{Q}_{g(X)}$), the constrained quadratic GAN would still have a fast convergence rate because only the empirical covariance matrix of $g(X)$ plays a role in its optimization which converges quickly to the population covariance matrix.

## 5 EXPERIMENTAL RESULTS

For experiments, we generate $n$ i.i.d. samples from $\mathbb{P}_Y \sim \mathcal{N}(\mathbf{0}, \mathbf{I}_d)$, represented as $Q_Y^n$. We then fit a $d$ dimensional Gaussian distribution $\mathcal{N}(\mu, \mathbf{K})$ to $\mathbb{Q}_Y^n$ using two methods: Maximum Likelihood (ML) estimation, which computes the sample mean and the empirical covariance; and WGAN (Arjovsky et al., 2017) with an affine generator function. Note that according to Theorem 3, ML

---

[2]Note that by considering $\psi(\mathbf{y}) = \mathbf{y}^t\mathbf{A}\mathbf{y}/2 + \mathbf{h}^t\mathbf{y}$ where $\mathbf{A} \succeq 0$, we would have obtained $Z \sim \mathcal{N}\left(\hat{\mu}_Y, \hat{\mathbf{K}}_Y\right)$ where $\hat{\mu}_Y$ is the sample mean. For simplicity, we ignore the affine terms.

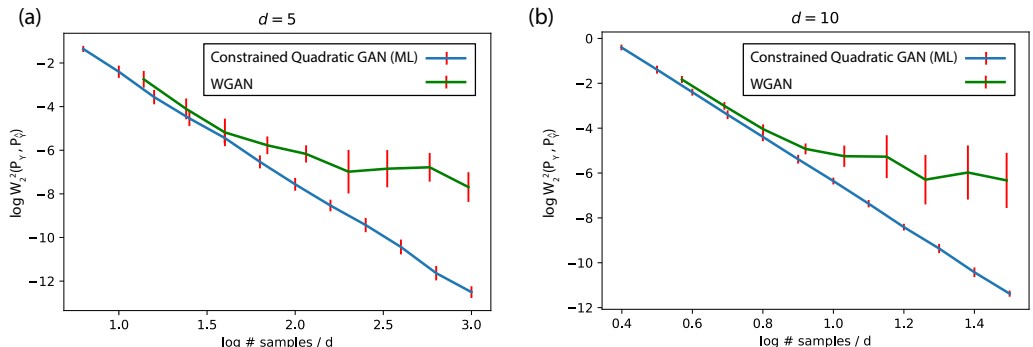

Figure 3: Generalization errors of constrained quadratic GAN (ML) and WGAN under the LQG setup.

is equivalent to the constrained quadratic GAN (19). Moreover, note that the WGAN implementation uses $W_1$ and not $W_2$ in its optimization. Although analyzing GANs with $W_2$ is more tractable than that of $W_1$, in practice we do not expect a significant difference between their performance. Considering this and owing to the lack of an implementation of GANs with $W_2$, we perform numerical experiments using the WGAN implementation. Details of the experiments can be found in Appendix A.

Let $\hat{\mu}$ and $\hat{\mathbf{K}}$ be the estimated mean and the covariance. For evaluation, we compute

$$\|\hat{\mu}\|^2 + \|\mathbf{I} - \hat{\mathbf{K}}^{1/2}\|_F^2, \tag{22}$$

which is the $W_2^2$ distance between $\mathcal{N}\left(\mathbf{0}, \mathbf{I}_d\right)$ and $\mathcal{N}\left(\hat{\mu}, \hat{\mathbf{K}}\right)$ (see Lemma 4 in Appendix B).

Figure 3 demonstrates the estimation error of the ML (constrained quadratic GAN) and WGAN methods for $d = 5$ and $d = 10$ and in different sample sizes. These figures are consistent with Theorem 2 and results of Section 3.2 which suggest that GAN's convergence can be slow owing to a bias in its optimal empirical solution with respect to the population one. Moreover, this figure shows that the convergence of the constrained quadratic GAN (ML) is fast. Finally, in our experimental results of Figure 3, one should take into the consideration practical issues of the WGAN implementation such as the use of the stochastic gradient descent, convergence to bad locals, etc.

## 6  DISCUSSIONS

From a broader perspective, the problem we addressed in this paper is that of finding a good generative model for data coming from a Gaussian ground-truth, $Y \sim \mathcal{N}(\mu, \mathbf{K})$. This is an age-old problem in statistics, and the baseline solution is using maximum likelihood estimation: one uses the data to estimate the mean and covariance matrix of the Gaussian distribution, i.e. the empirical mean $\hat{\mu}$ and empirical covariance matrix $\hat{\mathbf{K}}$, and obtain a generative model $\hat{Y} \sim \mathcal{N}(\hat{\mu}, \hat{\mathbf{K}})$. And when there is a desire to do dimensionality reduction, one can have a low-rank generative Gaussian model retaining the top $k$ principal components of $\hat{\mathbf{K}}$. This is the empirical PCA solution.

What we have shown in this paper is that there is a natural GAN architecture that can accomplish exactly these tasks (Figure 4).

While this is certainly a complicated way of performing maximum likelihood Gaussian estimation and PCA, we believe the result is interesting in several ways. First, it is not at all obvious that there is a natural GAN architecture that can accomplish this task. Since Gaussian modeling is a basic task, this is a good sanity check on GANs. Second, arriving at this GAN architecture requires us to make several advances in our understanding of GANs. We needed to find a general way to specify the loss function for GANs, and then specialize to the quadratic loss function for the Gaussian problem at hand. This led to the use of the second-order Wasserstein distance for the generator optimization, and to a general GAN architecture from the dual formulation of this optimization. We then needed to find a proper way to constrain the class of generators and the class of discriminators in a balanced

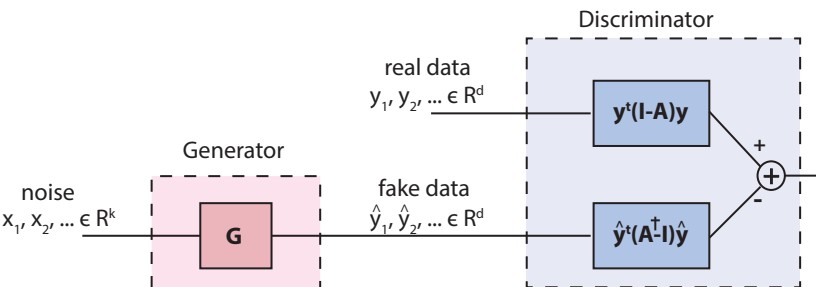

Figure 4: The GAN architecture that achieves maximum likelihood estimation for the zero-mean Gaussian model: a linear generator and a quadratic discriminator. On the training data, the generator minimizes over $G$ and the adversary maximizes over $A$.

way to achieve fast generalization. Indeed our goal was not to recover the maximum likelihood solution but to overcome the slow generalization when there are no constraints on the generator and the discriminator. That the final architecture with *balanced* generators and discriminators is also the maximum likelihood solution gives this story a satisfactory ending.

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

# Appendix

## A   DETAILS OF EXPERIMENTS

The WGAN is implemented in pytorch. Denote fully connected layer with the input dimension $d_{in}$ and the output dimension $d_{out}$ as $FC(d_{in}, d_{out})$. The generator can be represented as $FC(d, d)$; and the discriminator can be represented as $FC(d, n_f) - ReLU - FC(n_f, n_f) - ReLU - FC(n_f, n_f) - ReLU - FC(n_f, 1)$. The model is trained 100k iterations with batch size 128 with Adam optimizer. The learning rate is set to $2 \times 10^{-4}$. As for hyper parameters, $n_f$ is set to 128, the ratio of iterations between discriminator and generator is set to 5, and the weight clipping threshold is set to -0.02 and 0.02. Both ML and WGAN are repeated 10 times for each setting, and the mean and standard deviation is calculated and plotted (68.3% confidence interval).

## B   PROOFS

### B.1   NOTATION AND PRELIMINARY LEMMAS

For matrices we use bold-faced upper case letters, for vectors we use bold-faced lower case letters, and for scalars we use regular lower case letters. For example, $\mathbf{X}$ represents a matrix, $\mathbf{x}$ represents a vector, and $x$ represents a scalar number. $\mathbf{I}_n$ is the identity matrix of size $n \times n$. $\mathbf{1}_{n_1, n_2}$ is the all one matrix of size $n_1 \times n_2$. When no confusion arises, we drop the subscripts. $\mathbf{1}\{x = y\}$ is the indicator function which is equal to one if $x = y$, otherwise it is zero. $\text{Tr}(\mathbf{X})$ and $\mathbf{X}^t$ represent the trace and the transpose of the matrix $\mathbf{X}$, respectively. $\|\mathbf{x}\|_2 = \mathbf{x}^t \mathbf{x}$ is the second norm of the vector $\mathbf{x}$. When no confusion arises, we drop the subscript. $\|\mathbf{X}\|$ is the operator (spectral) norm of the matrix $\mathbf{X}$. $< \mathbf{x}, \mathbf{y} >$ is the inner product between vectors $\mathbf{x}$ and $\mathbf{y}$. $\mathbf{A}^\dagger$ is the pseudo inverse of the matrix $\mathbf{A}$. The eigen decomposition of the matrix $\mathbf{A} \in \mathbb{R}^{n \times n}$ is denoted by $\mathbf{A} = \sum_{i=1}^n \lambda_i(\mathbf{A}) \mathbf{u}_i(\mathbf{A}) \mathbf{u}_i(\mathbf{A})^t$, where $\lambda_i(\mathbf{A})$ is the $i$-th largest eigenvalue of the matrix $\mathbf{A}$ corresponding to the eigenvector $\mathbf{u}_i(\mathbf{A})$. We have $\lambda_1(\mathbf{A}) \geq \lambda_2(\mathbf{A}) \geq \cdots$. $\mathcal{N}(\mu, \mathbf{K})$ is the Gaussian distribution with mean $\mu$ and the covariance $\mathbf{K}$. $\text{KL}(\mathbb{P}_X, \mathbb{P}_Y)$ is the Kullback Leibler divergence between two distributions $\mathbb{P}_X$ and $\mathbb{P}_Y$. $\tilde{\mathcal{O}}(d)$ is the same as $\mathcal{O}(d \log(d))$.

**Lemma 4** *Let $Y \sim \mathcal{N}(\mathbf{0}, \mathbf{K}_Y)$ and $\hat{Y} \sim \mathcal{N}(\mathbf{0}, \mathbf{K}_{\hat{Y}})$. Then,*

$$W_2^2(\mathbb{P}_Y, \mathbb{P}_{\hat{Y}}) = Tr(\mathbf{K}_Y) + Tr(\mathbf{K}_{\hat{Y}}) - 2Tr\left( \left( \mathbf{K}_{\hat{Y}}^{1/2} \mathbf{K}_Y \mathbf{K}_{\hat{Y}}^{1/2} \right)^{1/2} \right) \tag{23}$$

$$= Tr(\mathbf{K}_Y) + Tr(\mathbf{K}_{\hat{Y}}) - 2Tr\left( \left( \mathbf{K}_Y^{1/2} \mathbf{K}_{\hat{Y}} \mathbf{K}_Y^{1/2} \right)^{1/2} \right).$$

**Proof 4** *See reference (Givens et al., 1984).*

### B.2   PROOF OF LEMMA 1

Let $Y = Y_{\mathcal{S}'} + Y_{\mathcal{S}}$ where $Y_{\mathcal{S}}$ represents the projection of $Y$ onto the subspace $\mathcal{S}$. Since

$$\mathbb{E}\left[ \|Y - \hat{Y}\|^2 \right] = \mathbb{E}\left[ \|Y_{\mathcal{S}'}\|^2 \right] + \mathbb{E}\left[ \|Y_{\mathcal{S}} - \hat{Y}\|^2 \right] \tag{24}$$

choosing $\hat{Y} = Y_{\mathcal{S}}$ achieves the minimum of optimization (9).

### B.3   PROOF OF THEOREM 1

Let $\mathcal{S}$ be a fixed subspace of rank $k$ where $\hat{Y}$ lies on. According to Lemma 1, if $\hat{Y}$ is unconstrained, the optimal $\hat{Y}^*$ is the projection of $Y$ onto $\mathcal{S}$ (i.e., $\hat{Y}^* = Y_{\mathcal{S}}$). Moreover, since $Y$ is Gaussian, $\hat{Y}$ is also Gaussian. Therefore, there exists a linear $g(.)$ such that $\mathbb{P}_{\hat{Y}^*} = \mathbb{P}_{g(X)}$ where $X \sim \mathcal{N}(\mathbf{0}, \mathbf{I})$. Thus, the problem simplifies to choosing a subspace where $\mathbb{E}\left[ \|Y_{\mathcal{S}}\|^2 \right]$ is maximized, which is the same as the PCA optimization.

## B.4 Proof of Theorem 2

Let $\mathbf{y}_1,...,\mathbf{y}_n$ be $n$ i.i.d. samples of $\mathbb{P}_Y$. Let $\hat{\mu}$ be the sample mean. Since $\mathbb{P}_Y$ is absolutely continuous, the optimal $W_2$ coupling between $\mathbb{Q}_Y^n$ and $\mathbb{P}_Y$ is deterministic (Villani, 2008). Thus, every point $\mathbf{y}_i$ is coupled with an optimal transport vornoi region with the centroid $\mathbf{c}_{y_i}^{(\mu,\mathbf{K})}$. Therefore, we have

$$W_2^2(\mathcal{N}(\mu,\mathbf{K}),\mathbb{Q}_Y^n) \tag{25}$$

$$= \|\mu\|^2 + \mathrm{Tr}(\mathbf{\Sigma}) + \frac{1}{N}\sum_{i=1}^N \|\mathbf{y}_i\|^2 - \frac{2}{N}\sum_{i=1}^N \mathbf{y}_i^t \mathbf{c}_{y_i}^{(\mu,\mathbf{K})}$$

$$= \|\mu\|^2 + \mathrm{Tr}(\mathbf{\Sigma}) + \frac{1}{N}\sum_{i=1}^N \|\mathbf{y}_i\|^2 - \frac{2}{N}\sum_{i=1}^N \mathbf{y}_i^t \left(\mathbf{U}\mathbf{\Sigma}^{1/2}\mathbf{U}^t \mathbf{c}_{y_i}^{(\mathbf{0},\mathbf{I})} + \mu\right)$$

$$= \|\mu\|^2 - 2\mu\hat{\mu} + \mathrm{Tr}(\mathbf{\Sigma}) + \frac{1}{N}\sum_{i=1}^N \|\mathbf{y}_i\|^2 - 2\mathrm{Tr}(\mathbf{U}\mathbf{\Sigma}^{1/2}\mathbf{U}^t\mathbf{A})$$

where

$$\mathbf{A} := \frac{1}{N}\sum_{i=1}^N \mathbf{c}_{y_i}^{(\mathbf{0},\mathbf{I})}\mathbf{y}_i^t. \tag{26}$$

The first step in (25) follows from the definition of $W_2$, the second step follows from the optimal coupling between $\mathcal{N}(\mu,\mathbf{K})$ and $\mathcal{N}(\mathbf{0},\mathbf{I})$, and the third step follows from the matrix trace equalities.

Therefore,

$$\triangledown_\mu W_2^2(\mathcal{N}(\mu,\mathbf{\Sigma}),\mathbb{Q}_Y^n) = 2\mu - 2\hat{\mu}, \tag{27}$$

which leads to $\mu_n^* = \hat{\mu}$. Moreover, each component of the sample mean is distributed according to $\mathcal{N}(\mathbf{0},1/n)$. Thus, $\|\mu_N^*\|^2 \sim \chi_d^2/n$ which goes to zero with the rate of $\tilde{\mathcal{O}}(d/n)$.

Let $\sigma_i^2$ be the $i$-th diagonal element of $\mathbf{\Sigma}$. Moreover, define $\mathbf{B} = \mathbf{U}^t\mathbf{A}\mathbf{U}$. Therefore, we have

$$\triangledown_{\sigma_i} W_2^2(\mathcal{N}(\mu,\mathbf{\Sigma}),\mathbb{Q}_Y^n) = 2\sigma_i - 2b_{i,i}, \tag{28}$$

where $b_{i,i}$ is the $i$-th diagonal element of the matrix $\mathbf{B}$. Thus, $\sigma_i^* = b_{i,i}$ and $\mathrm{Tr}(\mathbf{\Sigma}^*) = \mathrm{Tr}(\mathbf{B}) = \mathrm{Tr}(\mathbf{A})$.

Furthermore, we have

$$W_2^2(\mathbb{P}_Y,\mathbb{Q}_Y^n) = d + \frac{1}{N}\sum_{i=1}^N \|\mathbf{y}_i\|^2 - 2\mathrm{Tr}(\mathbf{A}), \tag{29}$$

which goes to zero with the rate of $\mathcal{O}(n^{-2/d})$ (Canas & Rosasco, 2012). Since $\frac{1}{N}\sum_{i=1}^N \|\mathbf{y}_i\|^2$ goes to $d$ with the rate of $\mathcal{O}(\sqrt{d/n})$ (because it has a $\chi$-squared distribution), $\mathrm{Tr}(\mathbf{A})$ goes to $d$ with a rate of $\mathcal{O}(n^{-2/d})$. Combining this result with $\mathrm{Tr}(\mathbf{\Sigma}^*) = \mathrm{Tr}(\mathbf{A})$ completes the proof.

