# OpenReview forum: "Understanding GANs: the LQG Setting"
_ICLR.cc/2018/Conference — Reject_

### Official Review · AnonReviewer1 · 2017-11-25
**GANs studied theoretically under extremely strong assumptions**

**Rating:** 4
**Confidence:** 4

**Review:**

*Paper summary*

The paper considers GANs from a theoretical point of view. The authors approach GANs from the 2-Wasserstein point of view and provide several insights for a very specific setting. In my point of view, the main novel contribution of the paper is to notice the following fact:

(*) It is well known that the 2-Wasserstein distance W2(PY,QY) between multivariate Gaussian PY and its empirical version QY scales as $n^{-2/d}$, i.e. converges very slow as the dimensionality of the space $d$ increases. In other words, QY is not such a good way to estimate PY in this setting. A somewhat better way is use a Gaussian distribution PZ with covariance matrix S computed as a sample covariance of QY. In this case W2(PY, PZ) scales as $\sqrt{d/n}$.

The paper introduces this observation in a very strange way within the context of GANs. Moreover, I think the final conclusion of the paper (Eq. 19) has a mistake, which makes it hard to see why (*) has any relation to GANs at all.

There are several other results presented in the paper regarding relation between PCA and the 2-Wasserstein minimization for Gaussian distributions (Lemma 1 & Theorem 1). This is indeed an interesting point, however the proof is almost trivial and I am not sure if this provides any significant contribution for the future research.

Overall, I think the paper contains several novel ideas, but its structure requires a *significant* rework and in the current form it is not ready for being published.

*Detailed comments*

In the first part of the paper (Section 2) the authors propose to use the optimal transport distance Wc(PY, g(PX)) between the data distribution PY (or its empirical version QY) and the model as the objective for GAN optimization. This idea is not novel: WGAN [1] proposed (and successfully implemented) to minimize the particular case of W1 distance by going through the dual form, [2] proposed to approach any Wc using auto-encoder reformulation of the primal (and also shoed that [5] is doing exactly W2 minimization), and [3] proposed the same using Sinkhorn algorithm. So this point does not seem to be novel.

The rest of the paper only considers 2-Wasserstein distance with Gaussian PY and Gaussian g(PX) (which I will abbreviate with R), which looks like an extremely limited scenario (and certainly has almost no connection to the applications of GANs).

Section 3 first establishes a relation between PCA and minimizing 2-Wasserstein distance for Gaussian distributions (Lemma 1, Theorem 1). Then the authors show that if R minimizes W2(PY, R) and QR minimizes W2(QY, QR) then the excess loss W2(PY, QR) - W2(PY, R) approaches zero at the rate $n^{-2/d}$ (both for linear and unconstrained generators). This result basically provides an upper bound showing that GANs need exponentially many samples to minimize W2 distance. I don't find these results novel, as they already appeared in [4] with a matching lower bound for the case of Gaussians (Theorem B.1 in Appendix can be modified easily to show this). As the authors note in the conclusion of Section 3, these results have little to do with GANs, as GANs are known to learn quite quickly (which contradicts the theory of Section 3).

Finally, in Section 4 the authors approach the same W2 problem from its dual form and notice that for the LQG model the optimal discriminator is quadratic. Based on this they reformulate the W2 minimization for LQG as the constrained optimization with respect to p.d. matrix A (Eq 16). The same conclusion does not work unfortunately for W2(QY, R), which is the real training objective of GANs. Theorem 3 shows that nevertheless, if we still constrain discriminator in the dual form of W2(QY, R) to be quadratic, the resulting soliton QR* performs the empirical PCA of Pn.

This leads to the final conclusion of the paper, which I think contains a mistake. In Eq 19 the first equation, according to the definitions of the authors, reads
\[
W2(PY, QR) = W2(PY, PZ),   (**)
\]
where QR is trained to minimize min_R W2(QY, R) and PZ is as defined in (*) in the beginning of these notes.
However, PZ is not the solution of min_R W2(QY, R) as the authors notice in the 2nd paragraph of page 8. Thus (**) is not true (at least, it is not proved in the current version of the text). PZ is a solution of min_R W2(QY, R) *where the discriminator is constrained to be quadratic*. This mismatch is especially strange, given the authors emphasize in the introduction that they provide bounds on divergences which are the same as used during the training (see 2nd paragraph on page 2) --- here the bound is on W2, but the empirical GAN actually does a regularized training (with constrained discriminator).

Finally, I don't think the experiments provide any convincing insights, because the authors use W1-minimization to illustrate properties of the W2. Essentially the authors say "we don't have a way to perform W2 minimization, so we rather do the W1 minimization and assume that these two are kind of similar".

* Other comments *
(1) Discussion in Section 2.1 seems to never play a role in the paper.
(2) Page 4: in p-Wasserstein distance, ||.|| does not need to be a Euclidean metric. It can be any metric.
(3) Lemma 2 seems to repeat the result from (Canas and Rosasco, 2012) as later cited by authors on page 7?
(4) It is not obvious how does Theorem 2 translate to the excess loss?
(5) Section 4. I am wondering how exactly the authors are going to compute the conjugate of the discriminator, given the discriminator most likely is a deep neural network?


[1] Arjovsky et al., Wasserstein GAN, 2017
[2] Bousquet et al, From optimal transport to generative modeling: the VEGAN cookbook, 2017
[3] Genevay et al., Learning Generative Models with Sinkhorn Divergences, 2017
[4] Arora et al, Generalization and equilibrium in GANs, 2017
[5] Makhazani et al., Adversarial Autoencoders, 2015

---

### Official Review · AnonReviewer2 · 2017-11-26
**This work does not explain GANs, it merely revisits minimum-distance density estimation**

**Rating:** 4
**Confidence:** 5

**Review:**

First of all, let me state this upfront: despite the sexy acronym "GAN" in the title, this paper does not provide any genuine understanding of GANs. Conceptually, GANs are an algorithmic instantiation of a classic idea in statistics, mamely minimum-distance estimation, originally introduced by Jacob Wolfowitz in 1957 (*). This provides the 'min' part. The 'max' part comes from considering distances that can be expressed as a supremum over a class of test functions. Again, this is not new -- for instance, empirical risk minimization, in both supervised and unsupervised learning, can be phrased as precisely such a minimax problem by casting the convergence analysis in terms of suprema of suitable empirical processes (see, e.g., "Empirical Processes in M-Estimation" by Sara Van De Geer). Moreover, even the minimax (and, more broadly, game-theoretic) criteria go back all the way to the foundational papers of Abraham Wald.

Now, the conceptual innovation of GANs is that this minimax formulation can be turned into a zero-sum game played by two algorithmic architectures, the generator and the discriminator. The generator proposes a model (which is assumed to be easy to sample from) and generates a sample starting from a fixed instrumental distribution; the discriminator evaluates the current proposal against a class of test functions, which, again, are assumed to be easily computable, e.g., by a neural net. One can also argue that the essence of GANs is precisely the architectural constraints on both the generator and the discriminator that make their respective problems amenable to 'differentiable' approaches, e.g., gradient descent/ascent with backpropagation. Without such a constraint, the saddle point is either trivial or reduces to finding a worst-case Bayes estimate, as classical statistical theory would predict.

This paper essentially strips away the essence of GANs and considers a stylized minimum-distance estimation problem, where both the target and the instrumental distributions are Gaussian, and the 'distance' between statistical models is the quadratic Wasserstein distance induced by the Euclidean norm. This, essentially, stacks the deck in favor of linear strategies, and it is not surprising at all that PCA emerges as the solution. It is very hard to see how any of this helps our understanding of either strengths or shortcomings of GANs (such as mode collapse or stability issues). Moreover, the discussion of supervised and unsupervised paradigms is utterly unconvincing, especially in light of the above comment on minimum-distance estimation underlying both of these paradigms. In either setting, a learning algorithm is obtained from the population version of the problem by substituting the empirical distribution of the observed data for the unknown population law.

Additional minor comments on proper attribution and novelty of results:

1) Lemma 3 (structural result for optimal transport with L_2 Wasserstein cost) is not due to Chernozhukov et al., it is a classic result in the theory of optimal transportation, in various forms due to Brenier, McCann, and others -- cf., e.g., Chapters 2 and  3 of C. Villani, "Topics in Optimal Transportation."

2) The rate-distortion formulation with fixed input and output marginal in Appendix A, while interesting, is also not new. Precise characterizations in terms of optimal transport are available, see, e.g., N. Saldi, T. Linder, and S. Yuksel, "Randomized Quantization and Source Coding With Constrained Output Distribution," IEEE Transactions on Information Theory, vol. 61, no. 1., pp. 91-106, January 2015.

(*) The method of Wolfowitz is not restricted to distance functions in the mathematical sense; it can work equally well with monotone functions of metrics -- e.g., the square of a metric.

---

### Official Review · AnonReviewer3 · 2017-12-12
**LQG setting reduces the GAN optimization to essentially PCA and I believe is too simple to give insights into more complex GANs.**

**Rating:** 5
**Confidence:** 4

**Review:**


Summary:
This paper studies GANs in the following LQG setting: Input data distribution (P_Y) is Gaussian with zero mean and Identity covariance. Loss function is quadratic. Generator is also considered to be a Gaussian distribution (linear function of the input Gaussian noise). The paper considers two settings for discriminator: 1)Unconstrained and 2) Quadratic function. For these settings, the paper studies the generalization error rates, or the gap between Wasserstein loss of the population version (P_Y) and the finite sample version (Q_n(Y)). The paper shows that when the discriminator is unconstrained, even though the generator is constrained to be linear, the convergence rates are exponentially slow in the dimension. However constraining the discriminator improves the rates to 1/\sqrt{#samples}. This is shown by establishing the equivalence of this setting to PCA.


Comments:


1) This paper studies the statistical aspects of GANs, essentially the sample complexity required for small generalization error, for the simpler LQG setting. The LQG setting reduces the GAN optimization to essentially PCA and I believe is too simple to give insights into more complex GANs.

2)The results show that using high capacity neural network architectures can result in having solutions with high variance/generalization error. However, it is known even for classification that neural networks used in practice have high capacity (https://arxiv.org/abs/1611.03530) yet generalize well on *real* tasks. So, having slow worst case convergence may not necessarily be an issue with higher capacity GANs, and this paper does not address this issue with the results.

3) The discussion on what is natural loss is very confusing and doesn't add to the results. While least squares loss is the simplest to study and generally offers good insights, I don't think it is either natural or the right loss to consider for GANs.

4) Also the connection to supervised learning seems very weak. In supervised learning generally Y is smaller dimensional compared to X, and generalization of g(X) depends on its ability to compress X, but still represent Y. On the contrary, in GANs, X is much smaller dimensional than Y.

---

### Public Comment · (anonymous) · 2017-10-30
**Modelling GANS via linear gaussians gone wrong.**

I came across this paper while searching for submissions on GAN theory. After going through this, my overall impression is that this paper doesn't analyze GANs but instead it trivially shows the solution to Least-Square under the Gaussian setting is PCA (a well known fact for more than 50 years).

Quick summary: This paper considers a GAN setting with a linear generator under squared error loss with Gaussian features. It shows that under the above assumptions, the optimal solution of the GAN is nothing but the PCA solution. The paper also has some basic simulations supporting Theorem 3 (the statement in the previous para). However, the paper has following major weakness:

1) One of the key drivers of NNs is the non-linearity between different layers. However, this paper restricts the generator to be linear, hence missing the key ingredient of NNs.

2) The main result says that GANs under second order Wasserstein loss is equivalent to PCA. However, in practice it is known that GANS are not doing PCA. In fact, GANS are able to achieve results superior than PCA on many datasets of interest. Isn't this a clear mismatch between practical observations and main conclusion of the paper itself? This raises the questions about the suitability of 'LQG' model itself.

3) The theory section of the paper motivates l_2  (2nd order wassertein) over the l_1 loss. However ironically,  the simulations use l_1 loss to justify the use of l2 of loss! Did I miss anything here?

4)  The paper doesn't use the fact that the generator and discriminator are NNs themselves. Thus the papre has nothing to with GANs as commonly understood by the community.

---

> ### Author Response · Authors · 2017-11-01
> **Understanding GANs via linear Gaussians is a good approach**
>
> Just two points of clarification about our approach to the problem:
>
> 1) There are two key aspects to GANs: i) the novel game theoretic learning architecture, ii) the use of neural networks as function classes over which the game theoretic objective is optimized. The complexity of understanding GANs stems from the combination of these two aspects. Our approach allows us to focus on aspect (i) by assuming a Gaussian data distribution. In this case, the class of linear generators is natural. Still it is not obvious what should be the class of discriminators for fast learning. Our result shows that one should use a class of quadratic discriminators to balance against linear generators. Thus our results are non-trivial and shed light on appropriate architectures for GANs even without bringing DNNs into the picture.
>
> 2) We want to emphasize we are NOT showing that Least-Square under the Gaussian setting is PCA. We start with proposing a general formulation for GANs with a general loss function and a general data distribution.  Then we show what happens when specializing this formulation to quadratic loss. The resulting problem is not least squares; it's minimizing the quadratic Wasserstein distance from the true distribution.  The connection with PCA is also more subtle. While in the population limit the solution is PCA, we show that without constraints on the discriminator the solution is NOT empirical PCA when there are finite many samples. Only when we put the quadratic constraint on the discriminator do we get back empirical PCA. Again, our results point to the importance of an appropriate constraining of the discriminator.

---

> > ### Public Comment · (anonymous) · 2017-11-04
> > **Modelling GANS via linear gaussians gone wrong**
> >
> > "(i) by assuming a Gaussian data distribution. In this case, the class of linear generators is natural " -
> >       Gaussian assumption on data might be reasonable but to analyze GANs linear generators are not natural assumption (as its the non-linearity that makes them powerful)
> >
> > (This) "shed light on appropriate architectures for GANs even without bringing DNNs into the picture"
> >       Simplifying GANs to linear (control) system with feedback removes the very essence GANs!
> >
> > "We start with proposing a general formulation for GANs with a general loss function and a general data distribution"
> >       Wasn't this proposed by Goodfellow et. al. ? Am I missing something?
> >
> > "The resulting problem is not least squares; it's minimizing the quadratic Wasserstein distance from the true distribution"
> >       Under the LGQ model, the paper solves :  min_g E[||Y-g*X||^2] for gaussian X, Y and linear operator g. Since the error term Y-g*X is gaussian , least square solution (obtained in the paper) is the MLE estimator (and MMSE).

---

> > > ### Author Response · Authors · 2017-11-05
> > > **Understanding GANs via linear Gaussians is a good approach**
> > >
> > > Response to the 4 comments:
> > >
> > > "Gaussian assumption on data might be reasonable but to analyze GANs linear generators are not natural assumption (as its the non-linearity that makes them powerful)"
> > >
> > > Our result showed the opposite. Without any constraints on the generator and allowing full nonlinearity, lemma 2 shows that the generalization ability of GAN is very poor, needing an exponential number of samples (exponential with the number of samples). With linear generators and quadratic discriminators, the generalization ability is much better, only linear number of  samples (Theorem 3)
> > >
> > > "Simplifying GANs to linear (control) system with feedback removes the very essence GANs!"
> > >
> > > We do not agree. As can be seen from the paper, there are lots of complexity in the problem even under the LQG setting.
> > >
> > > ""We start with proposing a general formulation for GANs with a general loss function and a general data distribution"  Wasn't this proposed by Goodfellow et. al. ? Am I missing something?"
> > >
> > > No the general formulation is not proposed by Goodfellow. For example, Wasserstein GANs are not covered by Goodfellow's formulation. Our formulation is a generalization of Wasserstein GAN's formulation.  (See Section 2)
> > >
> > > "Under the LGQ model, the paper solves :  min_g E[||Y-g*X||^2] for gaussian X, Y and linear operator g. Since the error term Y-g*X is gaussian , least square solution (obtained in the paper) is the MLE estimator (and MMSE). "
> > >
> > > No. min_g E[||Y - g*X||^2] is the supervised learning problem. The GAN problem  is unsupervised and is  given by
> > >
> > > min_g min_{P_X,Y} E [||Y - g*X||^2]
> > >
> > > Note that for the supervised learning problem, the empirical joint distribution p_X,Y is already given from the data. On the other hand, for the unsupervised learning problem, the joint distribution is not given and is part of the optimization. Hence a more complex problem. See Section 2 for more discussions.

---

### Public Comment · (anonymous) · 2017-11-11
**[Empirical conclusion] Not able to obtain convergence on squared W_2 loss**

Hi

As suggested to by paper's main result (but not shown experimentally), I tried training GANs with squared W_2 loss on LSUN-Bedrooms dataset over the last few days. However, my neural networks were not converging even after heavily tweaking hyper parameters (like learning rate, batch size etc).

Also, I was not able to find any suggestions in the longer version of this paper (from arxiv). I was wondering if you have any suggestions which could help me?

I tried using the following generator neural network models:
 1)  4 relu hidden layers and 512 units (as in Arjovsky's paper).
 2)  6 relu hidden layers and 512 units.
 3)  8 relu hidden layers and 256 units.

Thanks
Anna

---

> ### Author Response · Authors · 2017-11-18
> **About implementation**
>
> Hi Anna,
>
> Thank you for your interest in our work.
>
> Implementation of quadratic GAN (which has a convex discriminator) using deep neural nets (DNNs) is challenging due in part to the computation of the convex
> conjugate of the discriminator function \psi, which is a DNN. We are currently working on this part and haven’t extensively tested it.
>
> However, implementation of the method which we show that has a fast generalization rate in the LQG setup (quadratic GAN with convex quadratic discriminator) is easy since there is a closed-form solution for the convex conjugate of a convex quadratic function \psi.
>
> We are interested to know how are you implementing the quadratic GAN as the implementation details can affect the convergence.
>
> Best,
> On behalf of authors

---

> > ### Public Comment · (anonymous) · 2017-11-22
> > **Final Empirical conclusion**
> >
> > Dear Authors,
> >
> > Thanks for your reply.
> >
> > 1) I agree that computing the convex conjugate of a quadratic function \psi is easy (a very well known result indeed). However, this is cannot be used for implementation of a neural network. Also, one could directly use least squares if you want to stick to the linear setting.
> >
> > 2) Regarding my implementation, I was solving for \psi^* by gradient descent. What method did you try?
> >
> > Here are my final empirical conclusions:
> >
> > The only practical way to calculate psi^* is to perform many steps of gradient descent for *every* single step of GAN training and this completely destroys the computational efficiency of neural network.  Thus unfortunately this method is not practical by any measure. I wonder if this is the reason why this was not mentioned in "Wasserstein GAN" by Arjovsky et. al.

---

> > > ### Author Response · Authors · 2017-11-24
> > > **About implementation**
> > >
> > > Hi Anna,
> > >
> > > Thank you for your comment. Below are some clarifications:
> > >
> > > 1) Our main results show that without constraining the discriminator, quadratic GAN has a poor generalization even in the simple LQG setup. In this setup, we show that the proper constraint on the discriminator is the quadratic constraint which makes its convergence exponentially fast. Note that this constraint does not change the population limit compared to the unconstrained case (see Fig 1 in the paper). Indeed implementation of the quadratic GAN with quadratic discriminator is also straightforward.
> > >
> > > 2) In this work, we have not analyzed/implemented quadratic GAN  with neural network generators/discriminators. We believe that understanding GANs in the simple setup of the LQG paves the way to understand it (and properly implement it) in a more general setup. We agree with you that implementing the quadratic GAN with neural nets is more challenging than that of the WGAN (Arjovskey et al.) due in part to the computation of the convex conjugate. Note that if you use the gradient descent to implement this part, the extra running time should be comparable with the running time of optimizing the generator and the discriminator. Thus, even though the overall running time may be larger than that of current GAN implementations, it should be in the same order.

---

### Author Response · Authors · 2017-12-31
**Specific responses to reviews**

Some responses to specific comments by reviewers:

- Connection between supervised and unsupervised learning: This is a key step in our work. One reviewer seems to be saying that the connection is not new. However we cannot find a previous result on this. Can the reviewer give us a citation to a specific result? Another reviewer says that the connection is very weak because in a supervised setting typically the feature vector X is of a higher dimension than the target variable Y where in the GAN setting it is the reverse. We disagree with this reasoning. Take for example neural networks. They are invented in the supervised setting where the input X is typically of a higher dimension than the output Y. But when used as generators for GANs and also for autoencoders, the input is low dimensional and the output is high dimensional. So this already gives a hint of the connection between supervised and unsupervised learning. What we are doing in our paper is to make this connection explicit and at the problem formulation level rather than at the implementation level.

- The generalization distance Eq. (19): There seems a misunderstanding by Reviewer 2 who believes Eq. (19) contains a mistake. Actually the derivation is indeed correct although we think that the presentation can be improved. For clarification, let us elaborate the derivation as below. First of all, we emphasize that this derivation is w.r.t. the case where the discriminator is constrained to be quadratic (perhaps the misunderstanding on this misled the reviewer.) Since k=d, the optimal solution g*(X) for the population GAN matches the distribution of the real data Y, which yields: W_2^2(P_Y, P_{g*(X)}) = 0. This together with Eq. (10) then gives:
d_G(P_Y, Q_Y^n) = W_2^2(P_Y, P_{\hat{g}(X)}).
Here P_{\hat{g}(X)} indicates the optimal generated distribution w.r.t. “the case where the discriminator is constrained to be quadratic”. Hence, as per the derivation in Eqs. (17) & (18), P_{\hat{g}(X)} should be the same as the distribution generated by the sample covariance matrix (denoted by P_Z). This gives the one claimed in Eq. (19).

- “Neural networks have high capacity but can still generalize on real data, so worst case convergence results may not be relevant,” : Note that the poor generalization of GANs studied in this (and also in the paper by Arora et al) is due to the use of the Wasserstein distance measure in the inference. It is an orthogonal issue to the fact that neural networks have high capacity.

- Connection with N. Saldi, T. Linder, and S. Yuksel, "Randomized Quantization and Source Coding With Constrained Output Distribution,"  Thanks to the reviewer for the reference, which we didn't know of and will add in the revision. Theorem 7 in the reference is the result which is closest to Theorem 4 in our paper, if we set mu= psi=  P_Y. The function D(R) is the same as both cases.  However, W_2^2(P_Y, Q_Y^n) (with Q_Y_n = {y_1,.....y_n}  the data points randomly drawn from P_Y) in our paper is different from L_n(P_Y,P_Y, R) in the reference. In the language of randomized quantization, W_2^2(P_Y,Q_Y^n) is the minimum quantization error achieved with the constraint that each quantization point is equally likely under each realization of the random quantizer, while L_n(P_Y,P_Y,R) is the minimum quantization error achieved with the constraint that the overall distribution of the random quantizer's output matches P_Y. Since the latter constraint is looser, it can be seen that W_2(P_Y,Q_Y^n) \ge L_n(P_Y,P_Y,R). But it is not obvious that asymptotically they are equal. That's what we showed in our submission. However,  we find that to explain carefully this subtle but important difference will take us too far from the main thrust of the paper. So we have decided to remove the appendix and develop the material elsewhere.

---

### Author Response · Authors · 2017-12-31
**General responses to reviews**

Thank you to the reviewers for the detailed comments. We will try to state clearly what we believe is the main contribution of our submission and then use it to answer the main questions of the reviewers.

The driving question of our paper is that when the real data is Gaussian, what should be the natural GAN optimization problem? Our answer is

min_G max_{A psd}  \sum_i y_i^t (I-A) y_i - yy_i^t (A^\dagger - I)yy_i     ----------------(1)

Here y_i is the real data, yy_i = Gx_i is the fake data generated from the randomness x_i’s, and A^\dagger is the pseudo-inverse of the matrix A.

To make this contribution clear, we added a discussion section 6 in the revision. This optimization problem is highlighted in Figure 4.

1) Is this result novel? We believe so. We have never seen it in the literature.

2) Is this GAN? We believe so. It is a game played between the generator and a discriminator. The objective is a differentiable function of both G and A and so it is a specific algorithmic instantiation of the general abstract minimum distance estimation problem, tailored to Gaussian data.

3) Why are there no neural networks? They are not necessary for Gaussian data. The absence of neural networks is a consequence of our derivation, not an assumption.

4) Is the result too simple? We do not believe so. Even though there are no neural networks, the GAN objective function in (1) is far from trivial. While it is natural to have linear generators for a Gaussian problem, the specific form of the quadratic discriminator is not obvious and derived through a principled approach. In fact, there is a general belief in the field that the simplest discriminator is linear, but we show that even for a simple model like Gaussian data, linear discriminators do not suffice.

5) Is the Gaussian model relevant? Even though real world data is definitely more complex than Gaussian, it is a common and useful practice across many fields to understand a problem first in the Gaussian context (eg. linear regression, the Kalman filter). Then one can build on this result to extend to more general data distributions, introducing neural networks in the process (think about the evolution from linear regression to the perceptron to deep learning in supervised learning) The optimization problem (1) can serve as a useful concrete baseline for studying phenomena that would be too complex to first study in the general setting. For example, stability of training GANs, an important problem in general, can be studied first in the context of (1). If we do alternate gradient descent steps for G and for A, will we converge to the Nash equilibrium of the game? This is one among several follow up questions we are currently studying.

---

### Decision · Program_Chairs · 2018-01-29
**ICLR 2018 Conference Acceptance Decision**

**Decision:**

Reject

**Comment:**

While the reviewers agree that this is an important topic, there are numerous concerns novelty, correctness and limitations.